# In Silico Approach to Model Heat Distribution of Magnetic Hyperthermia in the Tumoral and Healthy Vascular Network Using Tumor-on-a-Chip to Evaluate Effective Therapy

**DOI:** 10.3390/pharmaceutics16091156

**Published:** 2024-08-31

**Authors:** Juan Matheus Munoz, Giovana Fontanella Pileggi, Mariana Penteado Nucci, Arielly da Hora Alves, Flavia Pedrini, Nicole Mastandrea Ennes do Valle, Javier Bustamante Mamani, Fernando Anselmo de Oliveira, Alexandre Tavares Lopes, Marcelo Nelson Páez Carreño, Lionel Fernel Gamarra

**Affiliations:** 1Hospital Israelita Albert Einstein, São Paulo 05652-000, Brazil; juanmunoz@usp.br (J.M.M.); giovana.fpileggi@einstein.br (G.F.P.); ariellydahora1997@gmail.com (A.d.H.A.); flavia.pedrini.ext@einstein.br (F.P.); nicolemev@gmail.com (N.M.E.d.V.); javierbm@einstein.br (J.B.M.); fernando.ao@einstein.br (F.A.d.O.); 2LIM44—Hospital das Clínicas da Faculdade Medicina, Universidade de São Paulo, São Paulo 05403-000, Brazil; mariana.nucci@hc.fm.usp.br; 3Departamento de Engenharia de Sistema Eletrônicos, Escola Politécnica, Universidade de São Paulo, São Paulo 05508-010, Brazil; alexandre.tavares@usp.br (A.T.L.); mnpcarreno@usp.br (M.N.P.C.)

**Keywords:** glioblastoma, magneto hyperthermia, magnetic nanoparticle, microfluidic device, heat dissipation

## Abstract

Glioblastoma multiforme (GBM) is the most severe form of brain cancer in adults, characterized by its complex vascular network that contributes to resistance to conventional therapies. Thermal therapies, such as magnetic hyperthermia (MHT), emerge as promising alternatives, using heat to selectively target tumor cells while minimizing damage to healthy tissues. The organ-on-a-chip can replicate this complex vascular network of GBM, allowing for detailed investigations of heat dissipation in MHT, while computational simulations refine treatment parameters. In this in silico study, tumor-on-a-chip models were used to optimize MHT therapy by comparing heat dissipation in normal and abnormal vascular networks, considering geometries, flow rates, and concentrations of magnetic nanoparticles (MNPs). In the high vascular complexity model, the maximum velocity was 19 times lower than in the normal vasculature model and 4 times lower than in the low-complexity tumor model, highlighting the influence of vascular complexity on velocity and temperature distribution. The MHT simulation showed greater heat intensity in the central region, with a flow rate of 1 µL/min and 0.5 mg/mL of MNPs being the best conditions to achieve the therapeutic temperature. The complex vasculature model had the lowest heat dissipation, reaching 44.15 °C, compared to 42.01 °C in the low-complexity model and 37.80 °C in the normal model. These results show that greater vascular complexity improves heat retention, making it essential to consider this heterogeneity to optimize MHT treatment. Therefore, for an efficient MHT process, it is necessary to simulate ideal blood flow and MNP conditions to ensure heat retention at the tumor site, considering its irregular vascularization and heat dissipation for effective destruction.

## 1. Introduction

Even today, brain tumors remain a significant challenge in the oncology field due to the patient’s poor survival rates and the heterogeneity of tumors [1]. Among them, glioblastoma multiforme (GBM) is the most frequent brain cancer in adults, comprising 49% of all cases, primarily in patients above 60 years [2]. Despite advancements in cancer therapeutics, GBM remains incurable, largely due to its infiltrative nature and resistance to conventional therapies [1]. GBM are among the most highly vascularized and edematous tumors, characterized by the overexpression of vascular endothelial growth factor (VEGF). This high VEGF expression drives angiogenesis, contributing to the extensive and abnormal vascular network typical of GBM [3,4,5,6]. Standard therapy typically includes maximal surgical resection, when possible, followed by a combination of radiotherapy and temozolomide chemotherapy [7]. However, the treatment of GBM is hindered by several factors, including insufficient drug delivery across the blood–brain barrier, significant intra- and intertumoral heterogeneity, signaling pathways that mediate tumor growth and tissue invasion, and an immunosuppressive microenvironment [8].

One crucial factor contributing to therapy resistance is the complex vascular network of GBM [9]. In contrast to the organized and structured vascular network of healthy tissue, intense angiogenesis in GBM results in the formation of a chaotic blood vessel network, characterized by tortuous pathways [10]. Functionally, these characteristics contribute to the irregular distribution of blood-borne therapeutic agents within the tumor microenvironment [11]. For this reason, the development of different therapeutic approaches, such as immunotherapy and nano-drug delivery systems, holds promising prospects [1]. Additionally, thermal therapies have emerged as potential treatments for GBM, including radiofrequency ablation, laser interstitial thermal therapy, photothermal therapy, and magneto hyperthermia (MHT). These techniques employ heat distribution to selectively target tumor cells, aiming to minimize damage to surrounding healthy tissues and overcome the limitations of conventional therapies [12,13].

MHT has gained attention for its noninvasive approach, heightened survival rates, and minimal side effects [14]. This therapy involves the use of an alternating magnetic field (AMF) to heat magnetic nanoparticles (MNPs) that are introduced into tumor tissues, as shown in Figure 1. The heating occurs through two main mechanisms: Néel relaxation and Brownian relaxation. In Néel relaxation, heat is generated by the reorientation of the magnetic moments of the nanoparticles, without physical movement, due to the overcoming of magnetic anisotropy barriers. In Brownian relaxation, heat is produced by the physical rotation of the nanoparticles in response to the AMF, creating friction with the surrounding fluid. Both mechanisms contribute to the controlled increase in temperature within the tumor tissue to a therapeutic level of around 42 °C, inducing cell lysis. The effectiveness of MHT depends on optimizing the characteristics of the MNPs, such as size and composition, and the AMF parameters, such as frequency and intensity. This approach allows for the selective heating of tumor areas, minimizing damage to normal tissues, and stands out as a promising technique in cancer therapies [15,16]. Due to the excessively sinuous blood vessels and the absence of lymphatic vessels, the low energy dissipation is amplified within GBM. This leads to sustained heating and causes tumor cell death, as these cells are more sensitive to temperature changes than healthy ones [8,17]. Thus, preclinical investigation of MHT therapy is crucial and relies on several methodologies, such as in vitro and in vivo models. In vivo tumor models using animal subjects offer valuable insights into biological mechanisms but fail to fully replicate human-specific features, resulting in low success rates in clinical trials and ethical concerns [18,19]. Furthermore, real-time monitoring of heat dissipation and the identification of potential tumor areas with insufficient temperature enhancement are not feasible in these models.

To address these limitations, organ-on-a-chip (OOAC) devices employ microfluidic technology to replicate the physiological microenvironment of particular tissues or organs, recapitulating their distinctive biophysics and biochemical features [20,21,22]. OOACs can be easily modified to meet specific objectives and can be designed to mimic the anomalous vascular network of GBM, providing critical insights into heat dissipation and other parameters that affect MHT therapy, such as particle size, shape, concentration, and temperature, as well as a flow rate that is similar to that found in capillaries and venules within a tumor [15,23,24,25,26,27]. Computational simulations can be integrated with these models, playing an important role in studying these devices and MHT parameters. In silico tools, such as COMSOL Multiphysics, enhance predictive accuracy and enable the definition of parameters that are difficult to measure experimentally [28,29,30,31,32]. These studies offer valuable information on physical phenomena, device viability, and the analysis of all parameters that can impact chip functionality and, consequently, the efficacy of the therapy [33].

Thus, this in silico study aims to use computational simulations conducted to optimize MHT therapy through two schematic diagrams of tumor-on-a-chip models, representing healthy and anomalous (GBM-inspired) vascular networks, which were designed to compare heat dissipation in continuous flow. These models provide insights into treatment strategies by predicting the effects of selected parameters on therapeutic outcomes. Additionally, they can be applied to evaluate other thermal therapies for treating tumors with similar vascular characteristics.

## 2. Materials and Methods

### 2.1. Microfluidic Devices Design Development to Evaluate Heat Dissipation in Vascular Networks

We developed two distinct vascular channel designs (healthy and abnormal vascularization of tumor) to simulate conditions in microchips for MHT analysis. The first design includes three linear channels, representing healthy vascularization. The second design consists of three interconnected complex channels, simulating the irregular, tortuosity, and heterogeneous distribution of anomalous vascularization observed in GBM cases. These designs were based on reference images [34] and detailed studies on the anatomy and microstructure of GBM vessels [35]. These models were built free-hand using the Autodesk^®^ Inventor (San Francisco, CA, USA) application version 2023, based on images documented in the literature, as shown in Figure 2A,B. Once the parameters required to achieve the therapeutic temperature are identified, we tested a vascular network with more complexicity vasculature, featuring channels of varying diameters and increased heterogeneity, closely resembling an in vivo tumor model. This approach was used to validate the hypothesis that the more complex and heterogeneous the vascular network, the faster the therapeutic temperature is reached.

The developed models were exported to COMSOL Multiphysics^®^ (Burlington, MA, USA) application version 6.1. To mimic an OOAC, the simulation regions were composed of polydimethylsiloxane (PDMS) in the outer domain, water in the mid-domain, and brain white matter in the inner domain. The PDMS structure, modeled as a hexahedron, measured 10 mm in width, 8 mm in length, and 1 mm in height. The water channel spanned 10 mm in width and 0.1 mm in height, while the brain white matter was modeled as a cylinder with a 0.4 mm radius and 0.08 mm height, as presented in Figure 2C,D.

The composition of the resulting model is predominantly PDMS, water, and white brain matter tissue, distributed as shown in Figure 2E,F, which are cross-sectional views of the 3D geometry in the region of z = 0.05 mm, equivalent to the intermediary region of the chip. Material properties for PDMS and water were obtained from the built-in COMSOL library, while those for brain white matter were based on the tissue database from IT’IS Foundation © (2010–2023) (Zurich, Switzerland) [36], and are summarized in Table 1.

After constructing the physical methods applied in the simulation, and defining the materials, geometries, and boundary conditions, the fine mesh predefined by COMSOL was selected, with the resulting mesh being visualized according to Figure 2G,H.

The water and PDMS materials are provided by the COMSOL library, with their respective parameters featuring more precise functions for a wider range of temperatures. However, the values presented in Table 1 are representative of the temperature range studied in the current study.

For the simulations, we employed the Fluid Flow Module with the Laminar Flow interface (spf) and the Heat Transfer Module with the Heat Transfer in Solids and Fluids interface (ht), as well as the Nonisothermal Flow interface in Multiphysics. Simulations were performed on a workstation equipped with an Intel(R) (Santa Clara, CA, USA) Xeon(R) Gold 5218R CPU, 191 GB of available RAM, and an NVIDIA (Santa Clara, CA, USA) RTX A4000 GPU.

### 2.2. Governing Flow Equations for Flow Evaluation in Microfluidic Devices

To understand the dependence of flow intensity on the heat dissipation mechanism, three different flow rates for a fixed concentration of 0.5 mg/mL of MNPs were evaluated, for 0, 1, and 100 µL/min. These rates were chosen to represent a wide range of conditions, including values below (0 µL/min), above (100 µL/min), and within the physiological range (1 µL/min) observed in tumor capillaries and venules [25,26,27].

The main equation governing the laminar flow interface is the momentum equation from the Navier–Stokes equations, expressed as:(1)$$\rho \frac{\partial u}{\partial t}+\rho (u\cdot \nabla )u=\nabla \cdot [-pI+\mu (\nabla u+(\nabla u)T)]+F$$
where *u* is the fluid velocity, *p* is the fluid pressure, *ρ* is the fluid density, *μ* is the fluid dynamic viscosity, and *F* are the external forces. In this context, the incompressibility of the flow is assured by *ρΔ⋅u* = 0, with initial values for the flow velocity and pressure set at *u* = 0 m/s and *p* = 0 Pa, respectively.

The forces that describe flow motion are presented in Equation (1), and each term of the differential equation represents:

$\rho \frac{\partial u}{\partial t}+\rho (u\cdot \nabla )u$: fluid flow related to its mass and initial velocity;

−∇⋅pI: spatial distribution of pressure;

$\nabla \cdot \left[\mu \left(\nabla u+\left(\nabla u\right)T\right)\right]$: dependence of fluid viscosity;

F: external forces.

### 2.3. Governing Heat Equations for Heat Dissipation Evaluation in Microfluidic Devices

Magneto hyperthermia functions are the main consideration of the heating process in the present study. Considering the flow as incompressible and the fluid as an ideal gas, we developed the governing equations for heat transfer, presented in Equation (2):(2)$$\rho C_{p}\frac{\partial T}{\partial t}+\rho C_{p}u\cdot \nabla T-\nabla (k\nabla T)=Q$$
where *C_p_* is the heat capacity at constant pressure, *T* is the temperature, *k* is the thermal conductivity, and *Q* is the total heat of the system. Boundary conditions for the heat transfer module were set with the entire system initially at 25 °C, an internal heat source in the tissue region described by Equation (1), all external walls as thermal insulators, and a convective heat flux at 25 °C coinciding with the fluid inlet and a heat outflow coinciding with the fluid outlet.

The forces that describe heat transfer are presented in Equation (2), and each term of the differential equation represents:

$\rho C_{p}\frac{\partial T}{\partial t}$: temperature variation for material density and specific heat;

$\rho C_{p}u\cdot \nabla T$: temperature spatial variation for flow domain;

−∇(k∇T): temperature variation due to the flow;

The heating function, obtained by MHT heating potency, related to the *Q* parameter in Equation (2), is described as:(3)$$P=\pi \mu_{0}\chi_{0}H^{2}f\cdot \frac{2\pi f\tau }{\left[1+{\left(2\pi f\tau \right)}^{2}\right]}$$

The magnetic field is proposed as a dependent function, acting only in Equation (3), and is described by Equation (4):(4)$$H=H_{0}\cdot cos(\omega t)$$

The heating time t ranged from 0 to 8000 s, at a 20 s’ step. The magnetic field H_0_ and frequency f were fixed at 23,427 kA/m and 420 kHz, respectively. The magnetic susceptibility is described as Equation (5):(5)$$\chi_{0}=\frac{\mu_{0}M_{s}^{2}V_{np}}{3k_{b}T}$$
where *M_s_* is the saturation magnetization, *V_np_* the volume of a single nanoparticle, and *k_b_* is the Boltzmann constant. To obtain the effective relaxation time, one needs to relate Neel and Brown relaxation times, obtained by Equations (6) and (7), respectively:(6)$$\tau_{N}=\tau_{0}\exp\left(\frac{KV_{np}}{k_{b}T}\right)$$
(7)$$\tau_{B}=\frac{3\eta_{nf}V_{h}}{k_{b}T}$$
where *K* is the anisotropy constant, *η_nf_* the nanofluid viscosity considering Fe_3_O_4_ nanoparticles and water, and *V_h_* is the hydrodynamic volume of nanoparticles. The relaxation time *τ_0_* was fixed in 10^−9^ s. Equation (8) presents the effective relaxation time:(8)$$\tau =\frac{\tau_{N}\cdot \tau_{B}}{\tau_{N}+\tau_{B}}$$

### 2.4. Main Parameters for Magnetic Nanoparticles and Concentration

To understand the dependence of MNP concentration on the heating rate mechanism, four different concentrations (0.25, 0.5, 0.75, and 1 mg/mL) were evaluated at a fixed flow rate of 1 µL/min, chosen to mimic the flow rate found in capillaries and venules within a tumor [25,26,27].

Considering that the tumor region is identical in both simulations, the distribution of MNPs within the tissue was considered uniform and monodisperse, since the study aims to evaluate heat dissipation due to the chip’s geometry.

The main parameters were developed based on considerations from previous studies on simulations of MHT [32,37], considering MNP diameter of 19 nm, density of 4.9 × 10^6^ g/m^3^, saturation magnetization *M_s_* of 446 A/m, and anisotropy constant *K* of 4.1 × 10^4^ J/m^3^.

## 3. Results

### 3.1. Evaluation of Geometry and Mesh Obtained after Processing the In Silico Study

The geometries were meticulously designed to simulate three distinct vascular models: model 1 represents normal vasculature, comprising 559,426 elements; model 2 simulates abnormal vasculature with a less complex network of tumor-associated vessels, comprising 2,673,254 elements; and model 3 depicts abnormal vasculature characterized by increased heterogeneity and complexity, consisting of 24,998,211 elements. These models were developed to evaluate heat dissipation in microfluidic devices during the analysis of the MHT process, with mesh parameters precisely tailored to meet the specific requirements of the study. For model 1, the mesh volume was 0.315 mm^3^, incorporating 559,426 elements, with an Average Element Quality (AEQ) of 0.6853 and a Minimum Element Quality (MEQ) of 0.0673, resulting in a processing time of approximately 2 h. Model 2, with a mesh volume of 0.516 mm^3^, contained 2,673,254 elements, an AEQ of 0.6737, and an MEQ of 0.0242, requiring about 4 h of processing time. Model 3, the most intricate, featured a mesh volume of 0.822 mm^3^, an AEQ of 0.6879, and an MEQ of 0.00105, with a processing time of approximately 24 h. Throughout the refinement process, channel corners were rounded to more accurately replicate the final microfluidic device structure, ensuring a faithful representation of the experimental conditions.

To enhance, understand and determine the parameters for MHT, we evaluated the influence of different flow rates on temperature distribution and velocity over time, as well as the effect of varying nanoparticle concentrations on temperature distribution over time. These assessments were conducted on only two models: the normal vasculature model and the low-complexity abnormal vasculature model. After determining the optimal parameters for MHT, these were applied to a third model with greater vascular complexity, closely resembling an in vivo model.

### 3.2. Evaluation of Velocity and Temperature Distribution for Different Inlet Flows in the Geometric Models of Microfluidic Devices

The evaluation of velocity distribution was conducted under two flow conditions: 1 µL/min (Figure 3A,B) and 100 µL/min (Figure 3C,D), showing a magnified view of the central region for each condition. For the flow rate of 1 µL/min, a maximum velocity of 8.1 × 10^−6^ m/s was observed in normal vasculature model 1 (Figure 3A). In contrast, the abnormal tumor vasculature model 2 showed a maximum velocity of 1.8 × 10^−6^ m/s (Figure 3B), 4.5 times smaller than the normal vasculature model. The maximum velocity values were predominantly located in areas that mimic tumor tissue. At the inlets and outlets, the average velocities were 5.2 × 10^−6^ m/s for the normal vasculature, and 0.91 × 10^−6^ m/s for the abnormal tumor vasculature, resulting in a velocity intensity ratio that was almost six times higher in the normal vasculature model.

For the flow rate of 100 µL/min, the maximum-recorded velocity was 7.9 × 10^−4^ m/s in normal vasculature model 1 (Figure 3C), while the abnormal tumor vasculature model 2 had a maximum velocity of 8.2 × 10^−5^ m/s (Figure 3D). These maximum velocities were also located near the tumor tissue, and in other channels at the inlets and outlets of the flow in abnormal vasculature model 2 as shown in Figure 3D. The average velocities at the inlets and outlets were 4.5 × 10^−4^ m/s for the normal vasculature, while the abnormal tumor vasculature model 2 exhibited variable velocity intensity, with a minimum value of 0.2 × 10^−5^ m/s. The velocity intensity ratio in normal vasculature model 1 was approximately ten times higher than in the abnormal tumor vasculature model 2.

For the vasculature models, there is also a difference in the magnitude of the point velocity values, around one order of magnitude, being higher for the healthy model. Additionally, while healthy model 1 shows consistent intensity in the developed channels, tumor model 2 shows significant variation in intensity in different channels, with regions of low hydrodynamic stress.

Analyzing the variability of the velocity map, a uniform distribution of intensities was observed in normal vasculature model 1, in contrast to the variability present in the abnormal tumor vasculature model 2. This difference in the uniformity of velocity distribution reflects the distinct characteristics between healthy and tumor vasculature (model 2), with important implications for the study of fluid dynamics in microfluidic devices and the efficacy of treatments based on MHT.

In addition, we evaluated the temperature map in both models and in the conditions of flow 0, 1, and 100 µL/min, during the simulation of the MHT process to investigate the temperature distribution around the chip on a cutting plane of z = 0.05 mm. This simulation was carried out considering a constant MNP concentration of 0.5 mg/mL (Figure 3E–P).

The evaluation of the heating process without flow (0 µL/min; Figure 3E–H) showed temperature maps in both models (1 and 2) with similar patterns of temperature increase, greater intensity in the central region of the models (tumor) due to not removing the heat generated by the interaction of the presence of MNPs in the tumor tissue and submission of the MHT heating process. Additionally, it is possible to observe heat dissipation outside the channels and the center of the models (1 and 2), given that the implemented system considers the existence of PDMS around the vascular network model. The inclusion of PDMS in this in silico study was for subsequent implementation in microfluidic devices and thus experimental evaluation.

Dead zones, in which there is no flow, but which correspond to the water material domain, assist in the heat dissipation process, considering that the thermal conductivity of water is greater than the thermal conductivity of PDMS. The walls of the PDMS act as a cold source, and it is possible to visualize an increase in temperature throughout the simulation geometry, indicating that the simulation does not concentrate the temperature variation only at the tumor site.

When evaluating the temperature map with flows 1 µL/min (Figure 3I–L) and 100 µL/min (Figure 3M–P), different temperature patterns were observed between the models (1 and 2), showing less heat retention at the tumor site with normal vasculature (Figure 3I,J,M,N) than with abnormal vasculature model 2 (Figure 3K,L,O,P). Due to the irregularity of the vessels that makes heat dissipation difficult, this behavior also occurs in heat dissipation around the tumor. Thus, removing this heat was more effective in normal vasculature model 1.

The temperature map also shows heat differences between the inlet and outlet of the flow. At the entrance (before arrival at the tumor), a lower temperature was observed than the region after the tumor (exit); this was due to the direction of flow of the liquid passing through the tumor region (heat source), thus leading to heat dissipation towards the tumor exit region.

Therefore, the temperature in the tumor is higher with lower flow (Figure 3I–L) and has less heat dissipation due to the irregularity of abnormal vasculature model 2 (Figure 3K,L).

### 3.3. Evaluation of Temperature Distribution for Different Inlet Flows over Time in the Geometric Models of Microfluidic Devices

In the analysis of temperature distribution over time in both vascularization models (1 and 2), we established three reference points located on the central horizontal axis at points X_1_ = 0 mm (black), X_2_ = 3 mm (red), and X_3_ = 6 mm (blue), as shown in Figure 4. The analysis cutoff time for each point was determined in tumor vasculature model 2 until the central area X2’s temperature reached 42 °C.

When evaluating the temperature with the flow equal to zero (Figure 4A,B), it was evident in both models (1 and 2) that heat dissipates isotropically when there is no flow influence and that the heating behavior in the MHT process is similar until reaching the therapeutic temperature of 42 °C, in X_2_ (corresponding to the tumor area) this occurred after 2500 s. Under this condition, there is neither a cooling source nor heat dissipation, and therefore, the heat generated in the tumor is not dissipated in either model, as shown in the inset of Figure 4A,B. As described in the previous topic, in the region corresponding to the inlet and outlet (X_1_ and X_3_, respectively) temperatures were lower than the tumor core (X_2_) over time, presenting similar values in both models (normal vasculature model 1: X_1_ = 40.61 °C; X_2_ = 42.17 °C; and X_3_ = 40.58 °C; and in abnormal vasculature model 2: X_1_ = 40.53 °C; X_2_ = 42.15 °C; and X_3_ = 40.53 °C).

For the analysis of the flow of 1 µL/min (Figure 4C,D), a difference of 5 °C in heating over time was observed between the models (1 and 2). In the tumoral abnormal vascular model 2, the tumor core (X_2_) reached the therapeutic temperature of 42.01 °C in 10,000 s. In the normal vascular model 1, at the same time and location, the temperature reached 37.82 °C, thus showing the influence of local heating as a function of flow (cooling source) and the geometry of the vasculature on heat dissipation in the system for the same heating source and input flow conditions, as shown the inset of Figure 4C,D. In the areas corresponding to the inlet (X_1_) and outlet (X_3_), lower temperatures were evident about the tumor site (X_2_), with the inlet temperature being lower than the outlet in both models (1 and 2) over time. However, the values in the normal vascularization model 1 were still lower than in the abnormal vascularization model 2, indicating an anisotropic heat dissipation process for studied dimensions in both models under pre-sent flow conditions. (Normal vasculature model 1: X_1_ = 34.86 °C; X_2_ = 37.82 °C; and X_3_ = 36,54 °C; and in abnormal vasculature model 2: X_1_ = 40.08 °C; X_2_ = 42.01 °C; and X_3_ = 39.02 °C.) Thus, under these flow conditions, it was possible to reach the therapeutic temperature only in the model of abnormal tumor vascularization.

At higher flow, such as 100 µL/min (Figure 4E,F), a similar flow behavior was observed as in the previous tests. However, at this increased flow rate (cooling source), heat dissipation was more efficient, as shown in the inset of Figure 4E,F. In abnormal vasculature model 2, the temperature in the tumor region reached only up to 40 °C in 8000 s, whereas in normal vasculature model 1 under the same conditions, the temperature in this region was 35 °C. Thus, high flow values allow for the removal of heat in tumor model 2 to the point of not reaching the therapeutic temperature. The temperature over time in the inlet (X_1_) and outlet (X_3_) regions was very similar to that described previously, presenting lower values than the tumor core region (X_2_), and the inlet was lower than the outlet in both models (1 and 2), as we can see in the following values, in normal vasculature model 1: X_1_ = 32.01 °C; X_2_ = 34.96 °C; and X_3_ = 33.87 °C; and in abnormal vasculature model 2: X_1_ = 36.31 °C; X_2_ = 39.10 °C; and X_3_ = 37.94 °C.

The temperature difference between flows of 1 and 100 µL/min was approximately 2 °C in both models (higher temperature for lower flow), and between the models in the X_2_ regions, the temperature increase for each flow was 5 °C (higher temperature for abnormal vasculature model 2), indicating that the flow intensity plays a crucial role in the heat dissipation of microfluidic systems for MHT.

Figure 4G,H shows the spatial distribution of temperature in a comparative way between the three analysis points (X_1_, X_2_, and X_3_) for the tested flows (0, 1, and 100 µL/min), showing progressive heating from the inlet (X_1_) up to a maximum in the center of the tumor (X_2_) and a slight decrease until outlet (X_3_).

### 3.4. Evaluation of Temperature Distribution over Time for Different Nanoparticle Concentrations in the Geometric Models of Microfluidic Devices

Considering the previous simulation, where the models (1 and 2) behaved adequately to evaluate the MHT process with a flow of 1 µL/min, in this topic we carried out the in silico evaluation of the heating and heat dissipation in both models as a function of some nanoparticle concentrations, such as 0.25, 0.5, 0.75 and 1 mg/mL. For this analysis of temperature variation over time, we used the same reference points as the previously described models, X_1_, X_2_, and X_3_.

At a concentration of 0.25 mg/mL, the temperatures reached in the healthy vascular model 1 (Figure 5A) at 8000 s were 30.03 °C for X_1_, 31.47 °C for X_2_, and 30.82 °C for X_3_. In tumor vascular model 2, the temperatures (Figure 5B) were slightly higher (31.92 °C for X_1_, 33.24 °C for X_2_, and 32.70 °C for X_3_). Therefore, at this concentration, it was not possible to reach, in both models (1 and 2), the therapeutic temperature (42 °C) in the MHT process due to the low concentration of MNPs, as can be confirmed in the inset images of both graphs (Figure 5A,B). The heating curves in X_1_ and X_3_ over time were smaller than at point X_2_ (tumor site), but with higher temperatures in X_3_ compared to X_1_.

At a concentration of 0.50 mg/mL (Figure 5C,D), tumor model 2 (Figure 5D) reached the therapeutic temperature in the tumor region (42.01 °C for X_2_) at 8000 s, and in other regions, it obtained a temperature of 39.12 °C for X_1_, and 40.93 °C for X_3_ while for healthy model 1 (Figure 5C), the temperatures reached were 34.91 °C for X1, 37.80 °C for X_2_, and 36.52 °C for X_3_. Therefore, with this concentration a temperature plateau pattern was observed for normal vasculature model 1, whereas in abnormal vasculature model 2, the temperature reached the therapeutic value, and with a slight increase trend, as can be confirmed in the inset images of both graphics (Figure 5C,D).

At concentrations of 0.75 (Figure 5E,F) and 1 mg/mL (Figure 5G,H), in tumor vasculature model 2, the reach of the therapeutic temperature (42 °C) was much faster, 2500 s and 1500 s, respectively, but in these same times healthy vasculature model 1 had a lower temperature reach (X_2_ = 40.01 °C and 40.92 °C, for the respective concentrations). Points X_1_ and X_3_ presented temperature values lower than X_2_ in both models (1 and 2), with X_1_ being lower than X_3_, which follows for the values in healthy vasculature model 1 (X_1_ (36.21 °C and 36.00 °C) and X_3_ (38.09 °C and 37.92 °C)) and in tumor model 2 (X_1_ (38.51 °C and 37.84 °C) and X_3_ (40.00 °C and 39.03 °C)), at the respective concentrations.

Analyses carried out with higher concentrations of MNPs reach the cutting temperature in a shorter treatment time (Figure 5). When comparing the specific temperature values in X_1_, X_2_, and X_3_, the highest values presented were always in X_2_, and in sequence X_3_. Next, we analyzed the differences in temperatures for the same points, comparing the models for the same concentrations of MNPs (0.25 mg/mL): ΔX_1_ = 1.89 °C, ΔX_2_ = 1.77 °C and ΔX_3_ = 1.88 °C; for 0.5 mg/mL: ΔX_1_ = 4.21 °C, ΔX_2_ = 4.21 °C and ΔX_3_ = 4.41 °C; for 0.75 mg/mL: ΔX_1_ = 2.30 °C, ΔX_2_ = 1.99 °C and ΔX_3_ = 1.91 °C; for 1 mg/mL: ΔX_1_ = 1.84 °C, ΔX_2_ = 1.08 °C and ΔX_3_ = 1.11 °C.

Therefore, in the process of MHT considering the vascularization present in tumor tissues, it is important to consider the following aspects as blood flow and abnormal vascular geometry present in the physiological system due to its relationship with the maintenance of local temperature and consequently the effectiveness of the therapy of MHT when using the ideal concentration of magnetic nanoparticle so that, subject to an oscillation of magnetic field and its intensity, it generates local heat for the destruction of tumor cells.

### 3.5. The Influence of Velocity and Temperature Distribution as a Function of Tumor Vascular Network Complexity in the MHT Process

To evaluate the influence of velocity and temperature distribution as a function of tumor vascular network complexity (Models 1, 2, and 3) in the MHT process, a flow rate of 1 µL/min, similar to that observed in capillaries and venules within tumors, was used along with a magnetic nanoparticle concentration of 0.5 mg/mL. These conditions were identified in previous results as the most effective for achieving the therapeutic temperature. Additionally, model 3 features a vascular network with varying channel diameters and increased heterogeneity, closely resembling an in vivo tumor model.

In the evaluation of velocity relative to vascular network complexity (different geometries; Figure 6A–C), the high-complexity vascular model 3 shows a maximum velocity of 0.42 × 10^−6^ m/s, which is approximately 19 times slower than in model 1 and 4 times slower than in model 2. Furthermore, the variability in velocity distribution significantly increases with the complexity of the vascular network, as clearly demonstrated in model 3 (Figure 6C), which closely resembles the vascular network of an in vivo tumor model.

In the evaluation of temperature distribution, as shown in Figure 6E,F, a decrease in heat dissipation is observed with increasing vascular complexity, particularly in the tumor region. Model 3 exhibited the lowest heat dissipation due to the tortuosity and varying diameters of the tumor vasculature, positively influencing the MHT process. Under the same conditions, the temperature reached after 10,000 s was 44.15 °C for model 3, 42.01 °C for model 2, and 37.80 °C for the normal model (Figure 6G,H).

Thus, the significant impact of vascular network complexity on velocity and temperature distribution during the MHT process can be observed. Model 3, with its complex and heterogeneous vasculature, achieved the highest therapeutic temperature, demonstrating that greater vascular complexity can improve heat retention. The results suggest that considering vascular heterogeneity is crucial for optimizing MHT treatment.

## 4. Discussion

The technique of MHT represents an alternative for cancer treatment by using MNPs to generate heat locally within tumors. [15,38]. This in silico study, utilizing organ-on-a-chip models to mimic normal and irregular vascular systems as found in tumors [39,40], was crucial to understanding how the specific vasculature of tumors, often more anomalous and dense than that of healthy tissues [41], influences heat dissipation. Understanding these differences optimizes the efficacy of MHT [42]. Moreover, the significance of this in silico study lies in its ability to identify parameters for the development of microfluidic devices, facilitating a more detailed and precise evaluation of MHT without relying on other platforms like in vivo studies [43,44].

In evaluating the velocity distribution in the two models (1 and 2), we can affirm that the observed differences under flow rates of 1 and 100 µL/min demonstrate the influence of the structural characteristics and geometries of normal and tumoral vasculatures, as shown in several literature studies. [26,27,32,45]. The normal vasculature presented significantly higher maximum and average velocities than the tumoral vasculature (model 2) under both flow conditions. Under a 1 µL/min flow, the maximum velocity in normal vasculature was 4.5 times higher than in tumoral vasculature, and this pattern persisted with a 100 µL/min flow, where the maximum velocity in normal vasculature was approximately 9.6 times higher. These differences indicate that tumoral vasculature has much greater resistance to flow due to its irregular and less organized structure [40]. While this resistance may negatively impact the distribution of nutrients and therapeutic agents, it is favorable for the MHT process. The lower flow velocity in the tumoral vasculature can result in greater heat retention, which is desirable for the efficacy of MHT [32]. The variability in velocity intensity in the tumoral model, with regions of low hydrodynamic stress, can increase heat concentration in tumoral areas [46,47], potentially improving treatment efficacy [40].

The evaluation of velocity across different vascular network complexities reveals that model 3, with its high complexity, exhibits significantly slower maximum velocities compared to Models 1 and 2. Specifically, the maximum velocity in model 3 is 19 times slower than in model 1 and 4 times slower than in model 2, indicating that increased vascular complexity leads to greater resistance to flow [32]. This slower velocity is likely due to the irregular and heterogeneous nature of the tumor vasculature, which creates more obstacles to blood flow. The variability in velocity distribution in model 3 suggests that the complex structure of the tumor vasculature not only hinders flow but may also contribute to localized areas of reduced hydrodynamic stress, potentially enhancing the retention of heat during MHT treatment [32,40].

The ability of more complex models, such as model 3, to reach higher therapeutic temperatures reflects the clinical reality that tumors with more disorganized vascular networks can be more responsive to thermal treatments due to how heat is retained and distributed. However, precise control of these temperatures is crucial, as values above the therapeutic range can cause undesirable damage to surrounding tissue. Therefore, it is essential that the alternating magnetic field equipment used in MHT be experimentally modulated to ensure that temperatures are maintained within the ideal therapeutic range, preventing tissue overheating and ensuring the safety and effectiveness of the treatment [8,48].

The analysis of temperature distribution in geometric models of microfluidic devices revealed significant differences between normal and tumoral vasculature, with important implications for the efficacy of MHT treatments. Evaluation of the heating process without flow (0 µL/min) showed that temperature maps and heating curves in both models (1 and 2) exhibited similar patterns of temperature increase, with higher intensity in the central region (tumor core). Without flow, the heat generated by MNPs is not efficiently removed, resulting in a central temperature increase to approximately 42 °C for both models. This approach is common in in vitro MHT studies [49,50,51,52,53], where the absence of flow leads to results that are not consistent with in vivo studies, complicating their application in translational research [54].

Considering a flow rate of 1 µL/min, which is similar to that found in capillaries and venules within a tumor, we observed that the tumoral vasculature (models 2 and 3) reached a maximum temperature of 42.01 and 44.15 °C [25,26,27]. This indicates that the irregular structure of tumor vessels retains more heat, allowing the MHT process to achieve therapeutic temperatures [15]. In contrast, the normal vasculature reached a maximum temperature of 37.80 °C under the same flow conditions, demonstrating more efficient heat dissipation, lower thermal retention, and, therefore, less effectiveness for the proposed therapeutic hyperthermia [32,55]. Greater vascular complexity of model 3 improves heat retention, especially in tumor regions. This indicates that the intricate structure of tumor vasculature, although challenging for traditional therapies, can be utilized to enhance the effectiveness of MHT. As a result, integrating vascular heterogeneity into the design of MHT protocols is crucial for optimizing treatment efficacy, ensuring that therapeutic temperatures are more effectively achieved and sustained in complex tumor environments [32,40].

When the flow rate was increased to 100 µL/min, the maximum temperatures recorded were 34.96 °C in normal vasculature (model 1) and 39.10 °C in tumoral vasculature (model 2). Although the tumoral vasculature still retained more heat, the increased flow facilitated more efficient heat removal, reducing the retained temperature in the system [56]. This suggests that higher flow rates can compromise the efficacy of MHT therapy, as the heat may not be sufficiently retained to reach the therapeutic temperature necessary to cause tumor cell lysis as described by Pennes [57], showing that increased blood flow leads to greater heat removal, reducing the retained temperature in the system [57,58,59].

These findings highlight the importance of personalizing flow conditions in tumor models to optimize the efficacy of MHT treatments. By adjusting flow parameters to more closely reflect the physiological conditions within tumors, we can significantly improve heat retention and, consequently, therapeutic efficacy, leveraging the irregular structure and higher flow resistance of tumor vasculature [32].

In the evaluation of temperature distribution over time for different nanoparticle concentrations in geometric models (1 and 2) of microfluidic devices, it was observed that with a concentration of 0.25 mg/mL of MNPs, the temperatures recorded at the tumor site were 31.47 °C in healthy vasculature model 1 and 33.24 °C in the tumoral model 2. These values are well below the therapeutic temperature of 42 °C necessary for tumor cell lysis. The low nanoparticle concentration resulted in insufficient heat generation, demonstrating that this MNP concentration in the MHT process is ineffective, necessitating the evaluation of higher concentrations [31,60].

When tested with a concentration of 0.50 mg/mL of MNPs, a significant improvement in heating was observed, especially at point X_2_ (tumor site). In the tumoral model 2, the temperature reached 42.01 °C, achieving the therapeutic temperature, while in healthy model 1, the temperature at point X_2_ was 37.80 °C. An intermediate concentration of MNPs can achieve the therapeutic temperature in tumor tissue effectively while ensuring safe temperatures in healthy tissues without causing damage. [61].

When higher concentrations of 0.75 and 1 mg/mL of MNPs were tested, the temperature at point X_2_ in the tumoral model 2 quickly reached the therapeutic temperature of 42 °C. In healthy model 1, the temperatures were 40.01 and 40.92 °C, correspondingly, for both concentrations. The rapid temperature increase in tumoral model 2 suggests that higher MNP concentrations are more effective in generating sufficient heat to reach the therapeutic temperature in a shorter time. However, this occurs in both the tumoral and healthy models (2 and 1, respectively). Therefore, it is important to ensure the concentration used is calibrated to heat tumor tissues without causing damage to healthy tissues [59,62]. The models (1 and 2) examined in this research enabled the assessment of the effect of MNP concentration employed in the MHT procedure.

The measured values before (X_1_) and after the tumor site (X_3_) showed consistent temperature behavior over time, with X_1_ always having a lower value than X_3_. This occurs because, as the liquid flows from X_1_ to X_3_, it absorbs heat from the X_2_ region (tumor area), which has a higher temperature concentration. The MNPs are distributed through the tissue and heated up according to the applied MHT process. As the liquid passes through X_2_ and X_3_, it continues to accumulate heat, increasing the temperature at X_3_. Thus, the higher temperature at X_3_ compared to X_1_ can be explained by heat accumulation along the flow path, higher heating efficiency near X_3_, and geometric and vascular factors that favor heat retention at X_3_ [32,45,55].

The analysis of the impact of vascular network complexity on the MHT process reveals important differences among the three models studied. Model 1, representing normal vasculature with uniform channels, showed the highest flow velocities and the lowest heat retention, indicating more efficient heat dissipation, which makes it more difficult to reach therapeutic temperatures. In model 2, simulating low-complexity tumor vasculature, a reduced flow velocity and greater heat retention were observed compared to model 1, reaching a temperature of 42.01 °C. This demonstrates how the structural irregularity of tumor vasculature begins to favor the MHT process. Model 3, simulating highly complex and heterogeneous tumor vasculature, exhibited the lowest flow velocity and the highest heat retention, reaching 44.15 °C. This shows that vascular complexity, with its tortuosity and variation in channel diameters, enhances the effectiveness of MHT by facilitating heat retention and the achievement of therapeutic temperatures.

These results highlight the importance of considering vascular heterogeneity in the planning of MHT treatments. While normal vasculature efficiently dissipates heat, more complex and disorganized tumor vascular networks favor better heat retention, making thermal treatment more effective. Understanding these differences is essential for developing more precise and effective approaches to treating tumors with MHT, by adjusting treatment parameters to the specific vascular architecture of each tumor.

Some limitations of the study include the use of a specific geometric model for microfluidic devices, which may not capture the full range of tumor vascular geometries. Additionally, only one size of MNP was considered, and future research could investigate how different sizes and shapes of MNP influence heat generation and treatment efficacy [63,64,65].

## 5. Conclusions

This study highlighted the central importance of vascular complexity in the effectiveness of magnetic hyperthermia therapy (MHT). By utilizing models based on microfluidic devices to accurately replicate normal and tumoral vascular systems, it became evident that the irregular and densely organized structure of tumor vasculature has a significant impact on heat retention and dissipation during MHT. The results show that, due to its higher resistance to blood flow, tumor vasculature is more effective at retaining heat compared to normal vasculature, facilitating the achievement of the necessary therapeutic temperatures. Consequently, lower flow rates in the tumor models proved particularly effective in heat retention, constituting a critical factor for the success of MHT.

The study also emphasized the importance of precisely adjusting the concentration of magnetic nanoparticles (MNPs) to ensure effective heat generation without compromising the integrity of healthy tissues. Additionally, the need to personalize treatment conditions to optimize MHT efficacy was highlighted. The proposed model demonstrated itself to be a robust tool for accurately simulating and assessing these variables, reducing the need for in vivo studies and overcoming the limitations of in vitro studies, which often do not consider the influence of vasculature and blood flow. This in silico study establishes a solid foundation for future research and advances in the clinical application of MHT, aiming for more effective cancer treatments.

In summary, this in silico approach provides a powerful tool for understanding how vascular geometry and blood flow dynamics influence MHT outcomes, underscoring the importance of tailoring treatment parameters to the specific vascular architecture of each tumor. This work lays a solid foundation for future research focused on optimizing MHT protocols and improving the clinical effectiveness of cancer therapies.

## Figures and Tables

**Figure 1 pharmaceutics-16-01156-f001:**
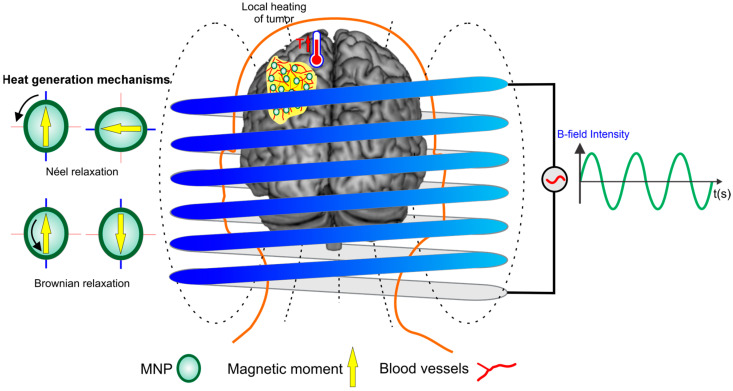
Representation of the therapeutic process of magneto hyperthermia in tumor treatment is based on the application of an alternating magnetic field in the presence of magnetic nanoparticles (MNPs). This process is governed by two primary heat generation mechanisms: Néel relaxation and Brownian relaxation.

**Figure 2 pharmaceutics-16-01156-f002:**
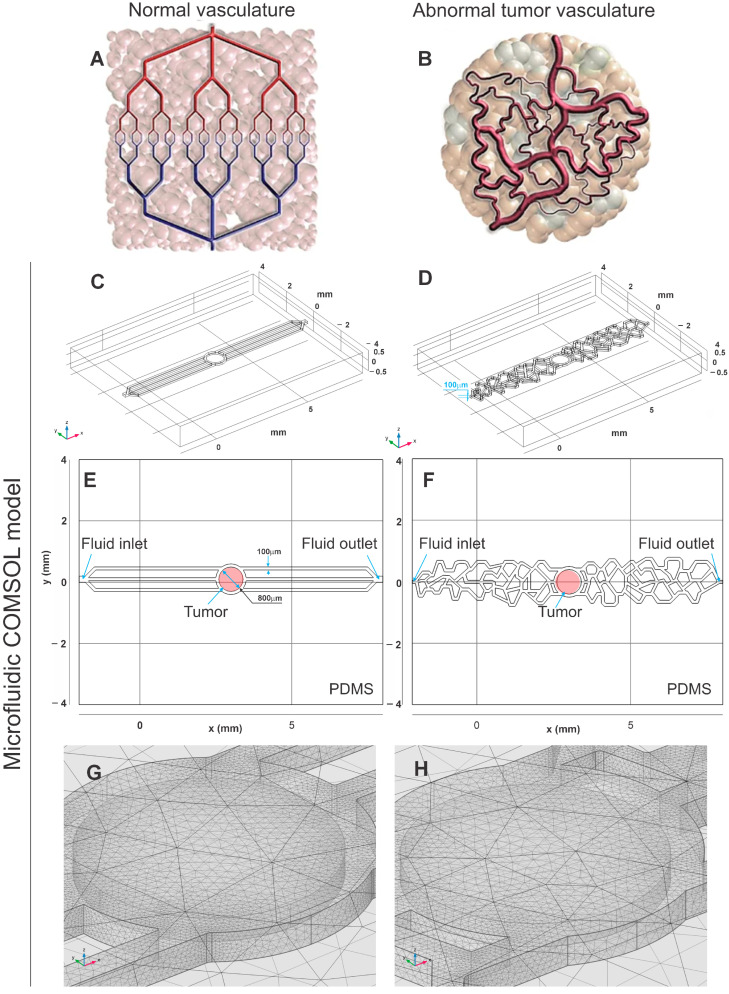
Representations of normal and abnormal tumor vasculature, respectively, (**A**,**B**) by diagram; (**C**,**D**) by three-dimensional geometry of the model in COMSOL; (**E**,**F**) by two-dimensional geometry of the model in COMSOL, with the regions defined; and (**G**,**H**) by the mesh used. ((**A**,**B**) Adapted with permission from [35].)

**Figure 3 pharmaceutics-16-01156-f003:**
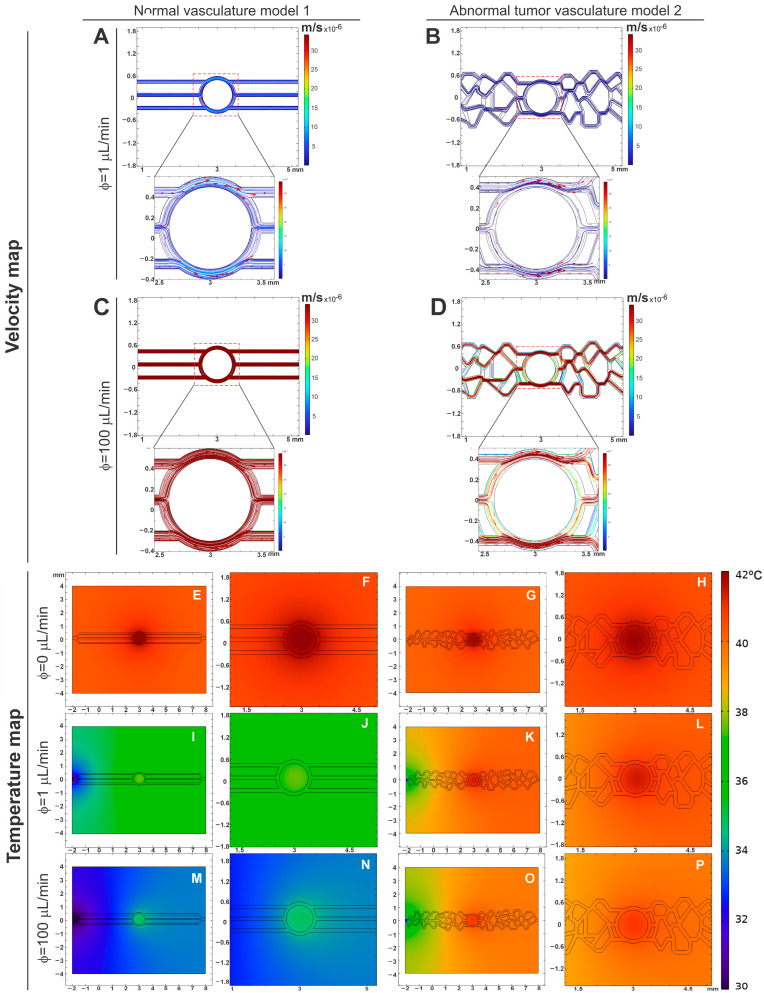
Maps of velocity (**A**–**D**) and temperature (**E**–**P**) for normal (model 1) and abnormal tumoral vasculature (model 2) under 0, 1, and 100 µL/min flow conditions.

**Figure 4 pharmaceutics-16-01156-f004:**
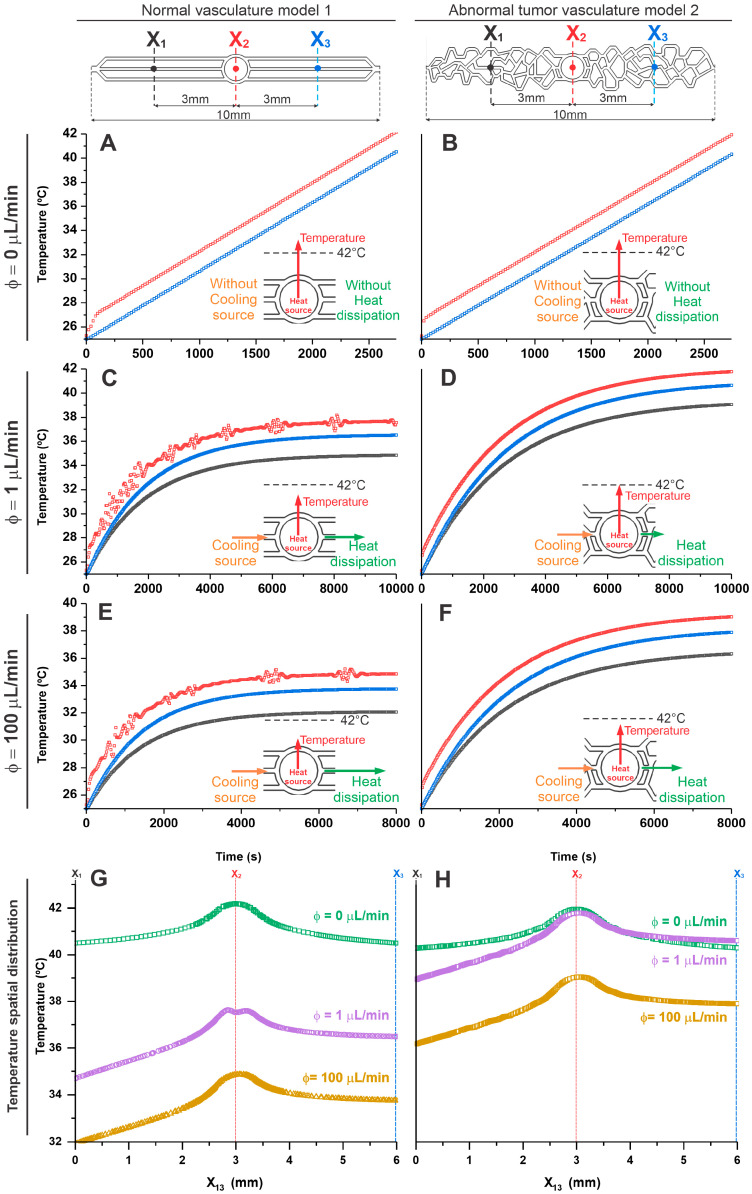
Evaluation of temperature distribution over time for different inlet flows, 0 µL/min (**A**,**B**), 1 µL/min (**C**,**D**), and 100 µL/min (**E**,**F**) in the geometric models of normal (model 1—the left graphics) and abnormal vasculature (model 2—the right graphics) in the selection regions X_1_ (black lines), X_2_ (red lines), and X_3_ (blue lines). (**G**,**H**) The temperature spatial distribution shows the comparison between the selection regions for the tested flows (0, 1, and 100 µL/min) in both models. The insets of each graph represent the heat balance resulting from the cooling source (orange color), heat source (red color), and heat dissipation (green color) for each condition.

**Figure 5 pharmaceutics-16-01156-f005:**
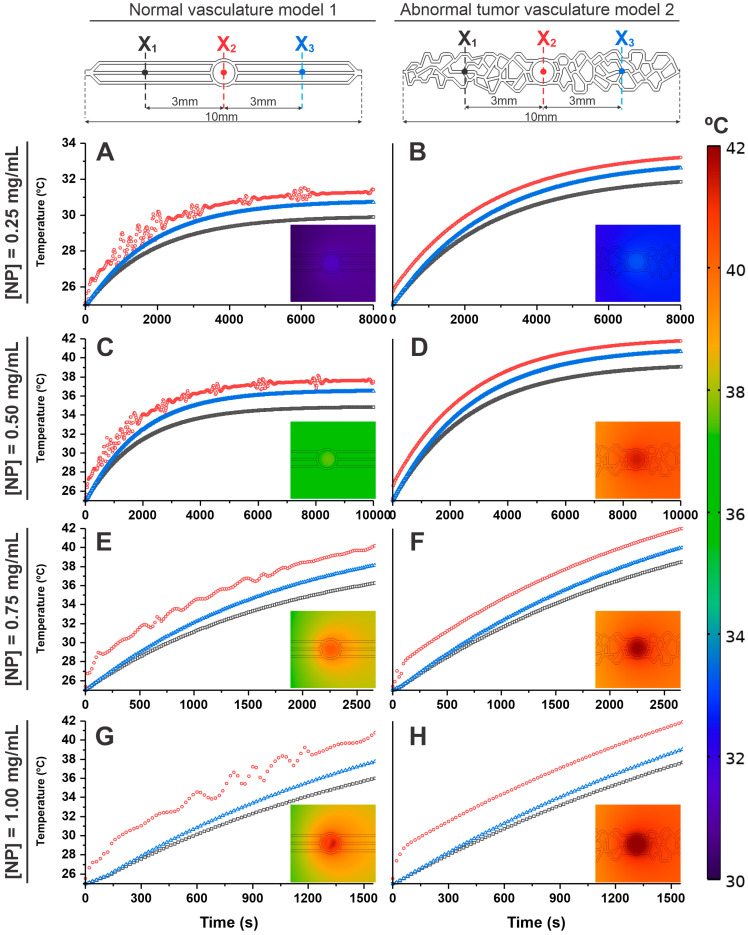
Evaluation of temperature distribution over time for 1 µL/min of flow, considering different nanoparticle concentrations, (**A**,**B**) 0.25 mg/mL, (**C**,**D**) 0.50 mg/mL, (**E**,**F**) 0.75 mg/mL, and (**G**,**H**) 1.00 mg/mL in both geometric models, in the right side, normal vasculature model 1, and the left side the abnormal tumoral vasculature model 2, in the selection regions X_1_ (black lines), X_2_ (red lines), and X_3_ (blue lines).

**Figure 6 pharmaceutics-16-01156-f006:**
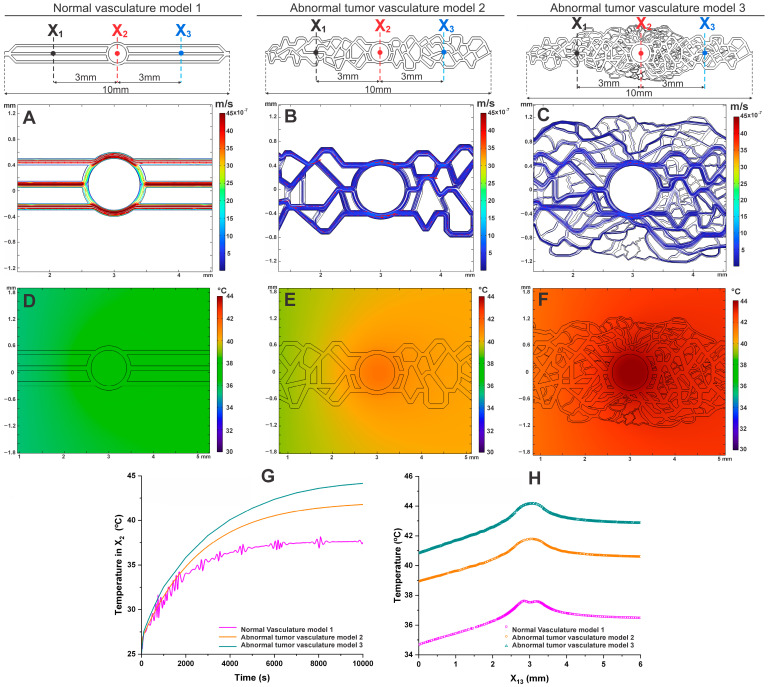
Velocity (**A**–**C**) and temperature (**D**–**F**) maps in vascular networks, evaluating the influence of velocity and temperature distribution as a function of tumor vascular network complexity in the MHT process. Temperature measurements in the tumor tissue (X2) up to 10,000 s (**G**), with a spatial temperature profile in the tumor (**H**).

**Table 1 pharmaceutics-16-01156-t001:** Representative materials parameters used in the simulation for a temperature range of 25 to 42 °C.

Parameter	Materials		
	PDMS	Water	White Matter
Density [kg/m^3^]	970	998	1.04 × 10^3^
Thermal conductivity [W/(m∙K)]	0.16	0.60	0.48
Heat capacity at constant pressure [J/(kg∙K)]	1.46 × 10^3^	4.18 × 10^3^	3.58 × 10^3^
Dynamic viscosity [Pa∙s]	--	7.71 × 10^−4^	--
Fluid inlet [m/s]	--	8.49 × 10^−6^	--

## Data Availability

The original contributions presented in the study are included in the article, further inquiries can be directed to the corresponding author.

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
