# Peer review of "In Silico Approach to Model Heat Distribution of Magnetic Hyperthermia in the Tumoral and Healthy Vascular Network Using Tumor-on-a-Chip to Evaluate Effective Therapy"

_pharmaceutics, 2024, doi:10.3390/pharmaceutics16091156_

Round 1

Reviewer 1 Report

Comments and Suggestions for Authors

The authors have used in-silico approach to assess the usage of magnetic hyperthermia to treat glioblastoma multiform. This is a good approach and use of numerical methods to study the physical process before experiments and clinical trials.

The authors need to clarify certain assumptions and results before publication:

1. The vacular networks used to model normal and tumor vasculatures are not inspired from any iv-vivo images. This makes the normal vasculature to be very simple and engineered. By assuming the normal vasculature to be like a tree, mostly used for illustration it does not fully recapitulate the native vascular network. It would be better to include networks that are derived from the images of normal and tumor vasculatures.

2. The authors find the velocities near the tumors to be different for the two models. Can this be explained in depth as this is a counter intuitive observation? When the inlet boundary conditions are same, there is no dissipation of fluid in the network and the vascular dimensions are the same near the tumor, the velocities should be the same for the conservation of mass. This may have consequences on the conclusions of this study.

3. Figure 2, A_D needs the same scale bar for the velocity heat map, which would make it easier for the readers to understand the figure

Author Response

Reviewer #1

The authors have used in-silico approach to assess the usage of magnetic hyperthermia to treat glioblastoma multiform. This is a good approach and use of numerical methods to study the physical process before experiments and clinical trials.

The authors need to clarify certain assumptions and results before publication:

  1. The vascular networks used to model normal and tumor vasculatures are not inspired from any iv-vivo images. This makes the normal vasculature to be very simple and engineered. By assuming the normal vasculature to be like a tree, mostly used for illustration it does not fully recapitulate the native vascular network. It would be better to include networks that are derived from the images of normal and tumor vasculatures.

Answer: We appreciate the valuable feedback and suggestions. Our study aimed to develop a simplified model using microfluidic devices to evaluate thermal therapies, emphasizing understanding the fundamental mechanisms in a controlled environment. We modeled a simplified normal vascular network and a moderately complex, tortuous tumor vascular network to capture the basic characteristics of these structures.

The choice of these simplified, tree-like structures for normal vasculature was motivated by the need to create a clear and illustrative comparison between normal and tumor vasculatures, establishing a consistent reference baseline for analysis. However, we recognize the importance of using models that more accurately replicate physiological conditions. We plan to incorporate these improvements in future studies as we refine our research methodologies.

Our primary goal was to provide a proof of concept for the use of microfluidic devices as an innovative alternative to animal models in the evaluation of thermal therapies. This approach allows for controlled experimentation and represents a novel methodology, as there are currently no devices described in the literature that evaluate thermal processes alongside vascularization analysis.

We also recognize the importance of using vascular networks derived from in vivo images for more accurate simulations and plan to incorporate these networks in future studies. As we refine our model, we will increase the complexity of the vascular network to more closely mimic native tissue environments, allowing for a more comprehensive evaluation of therapeutic effects.

Additionally, this limitation was mentioned in the discussion section of the manuscript, acknowledging the need for more complex models in future investigations. We appreciate the suggestion and look forward to enhancing our model in future work to better recapitulate native vascular conditions both in silico and experimentally.

  1. The authors find the velocities near the tumors to be different for the two models. Can this be exp{Doutel, 2021 #54}lained in depth as this is a counter intuitive observation? When the inlet boundary conditions are same, there is no dissipation of fluid in the network and the vascular dimensions are the same near the tumor, the velocities should be the same for the conservation of mass. This may have consequences on the conclusions of this study.

Answer:  Thank you for your suggestion Considering the heterogeneity of the vessels present in the tumor, the design of the microfluidic device included inlets with different geometries to simulate this aspect, resulting in variations in velocity, although the flow is maintained constant at the inlet of the device for both models. In microfluidic systems, flow control is a crucial parameter, and in this platform, the variation in the geometry of the inlets reflects the heterogeneity of the vessels, generating variability in the inlet velocities between the models, thereby mimicking the tumor behavior. It is important to highlight that considering the same geometry at the inlet does not mimic the heterogeneity of the tumor tissue. We can assert that the flow at the inlet of the microfluidic device will be equivalent to the flow around the tumor in both models.

  1. Figure 2, A_D needs the same scale bar for the velocity heat map, which would make it easier for the readers to understand the figure

Answer:  Thank you for your suggestion. We have updated Figure 2 to include the same scale bar across all subfigures A-D, making it easier to compare the velocities. This change is reflected in the new version of Figure 3, as the previous Figure 2 has been renumbered due to the inclusion of an additional figure suggested by one of the reviewers.

Reviewer 2 Report

Comments and Suggestions for Authors

To validate the therapeutic efficacy of magnetic hyperthermia (MHT), Munoz et al. proposed an organ-on-a-chip model designed to emulate the intricate vascular network of glioblastoma multiforme (GBM), a type of brain cancer. They systematically simulated the effects of flow rate, microchannel architecture, and concentrations of magnetic nanoparticles on temperature distribution. The methodology is logically sound, and the paper provides sufficient experimental detail. The manuscript could be accepted for publication after addressing the following points:

The flow rate should be selected based on physiological conditions in vivo, such as that in brain tissue, to ensure the clinical relevance of the study is accurately reflected.

Author Response

Reviewer #2

o validate the therapeutic efficacy of magnetic hyperthermia (MHT), Munoz et al. proposed an organ-on-a-chip model designed to emulate the intricate vascular network of glioblastoma multiforme (GBM), a type of brain cancer. They systematically simulated the effects of flow rate, microchannel architecture, and concentrations of magnetic nanoparticles on temperature distribution. The methodology is logically sound, and the paper provides sufficient experimental detail. The manuscript could be accepted for publication after addressing the following points:

  1. The flow rate should be selected based on physiological conditions in vivo, such as that in brain tissue, to ensure the clinical relevance of the study is accurately reflected.

Answer:  Thank you for your suggestion. In this study, we selected flow rates based on those found in capillaries and venules within a tumor to ensure the clinical relevance of the study. To this end, we selected three flow conditions: 0, 1, and 100 µL/min. These rates were chosen to represent a wide range of conditions, including values below (0 µL/min), above (100 µL/min), and within the physiological range (1 µL/min) observed in tumor capillaries and venules. The results demonstrated that the proposed models were favorable for the in vivo physiological condition at the flow rate of 1 µL/min.

We included a new sentence in the introduction, with the appropriate references, and in the methodology (items 2.2 and 2.4) to describe the values used for both MNP concentration and the applied flow rates, covering conditions below, above, and within the range that mimics the flow found in capillaries and venules within a tumor. In the results and discussion of the initial manuscript, the findings based on these physiological conditions have already been described.

Reviewer 3 Report

Comments and Suggestions for Authors

The manuscript reports an heat dissipation model of magnetic hyperthermia (MHT) in microfluidic device by mimicking vascular networks, which is a feature of tumor environment like Glioblastoma multiforme (GBM). 

(1) Authors are recommended to present a schematic picture to introduce the principle of MHT, in particular using magnetic nanoparticle (MNP). Even if the general introduction of MHT is described in the introduction part, a standalone figure is needed for the general readers.

(2) Also, to understand the authors' model about heat balance, a simple schematic figure is needed to understand the heat flow from the oscillated MNP in the localized tumor site to be dissipated by the flow of vasculature: A picture description of heat dissipation is needed with total heat (Q) by magnetic oscillation and temperature distribution (T) by fluid flow (u). 

(3) I think if the authors have a histology picture with VEGF staining in GBM sample, it would be great to present it to show the complexity of vascular network in GBM. 

Author Response

Reviewer #3

The manuscript reports an heat dissipation model of magnetic hyperthermia (MHT) in microfluidic device by mimicking vascular networks, which is a feature of tumor environment like Glioblastoma multiforme (GBM). 

(1) Authors are recommended to present a schematic picture to introduce the principle of MHT, in particular using magnetic nanoparticle (MNP). Even if the general introduction of MHT is described in the introduction part, a standalone figure is needed for the general readers.

Answer: Thank you for your valuable suggestion. We concur that the inclusion of a schematic image illustrating magneto hyperthermia therapy in the introduction will significantly enhance the readers' comprehension of the technique, thereby aiding in a clearer understanding of the study's objective. Consequently, we have incorporated Figure 1 in the introduction, accompanied by a descriptive caption.

(2) Also, to understand the authors' model about heat balance, a simple schematic figure is needed to understand the heat flow from the oscillated MNP in the localized tumor site to be dissipated by the flow of vasculature: A picture description of heat dissipation is needed with total heat (Q) by magnetic oscillation and temperature distribution (T) by fluid flow (u). Answer:  Thank you for your suggestion. We have included a schematic inset in each graph of Figure 4 to enhance the understanding of the heat balance about flow, local tumor temperature influenced by the presence of MNPs, and heat dissipation, with the relevant quantities represented by arrows. Additionally, an explanation of the heat balance has been added to the corresponding results section.

(3) I think if the authors have a histology picture with VEGF staining in GBM sample, it would be great to present it to show the complexity of vascular network in GBM. 

Answer:  Thank you very much for the suggestion. Unfortunately, we do not have the histological figure with VEGF staining in GBM samples to include in the manuscript to illustrate the complexity of the vascular network in GBM. However, we have incorporated more information and citations from studies that demonstrate this complexity in the introduction section of the manuscript.

Round 2

Reviewer 1 Report

Comments and Suggestions for Authors

The authors have added more verbatim and explanations in text and figures to clarify the manuscript for the readers. 

However, the central hypothesis that tumor vasculature is abnormal and has a slower flow rate and lower heat dissipation is not corroborated by using an artificial network for control. The current tumor vasculature seems more like a control network, and the control network is similar to most of the networks used for illustration purposes in review articles. If the authors can add one more network architecture that involves a control network similar to in vivo architecture, this manuscript would be highly improved. Also, the authors central hypothesis will also be supported by the fact that most of the tumor vasculatures are more leaky than control networks. So, in the current form there are gaps in the manuscript that may not add value to the field and hence it is highly recommended to add one more figure panel that contains in vivo architecture of the normal and tumor vasculature and their corresponding performance.

Author Response

Reviewer #1

The authors have added more verbatim and explanations in text and figures to clarify the manuscript for the readers.

However, the central hypothesis that tumor vasculature is abnormal and has a slower flow rate and lower heat dissipation is not corroborated by using an artificial network for control. The current tumor vasculature seems more like a control network, and the control network is similar to most of the networks used for illustration purposes in review articles. If the authors can add one more network architecture that involves a control network similar to in vivo architecture, this manuscript would be highly improved. Also, the authors central hypothesis will also be supported by the fact that most of the tumor vasculatures are more leaky than control networks. So, in the current form there are gaps in the manuscript that may not add value to the field and hence it is highly recommended to add one more figure panel that contains in vivo architecture of the normal and tumor vasculature and their corresponding performance.

Answer:  We appreciate the reviewer’s valuable suggestion, which led us to incorporate a new network architecture into the revised manuscript that more accurately simulates the in vivo vasculature, both normal and tumoral. This new network was compared with the previously used ones, and the results were included in a new figure (Figure 6), demonstrating the differences in heat dissipation and flow dynamics between the tumoral and normal vasculatures. These findings reinforce the central hypothesis of the manuscript. Additionally, the text was updated to reflect these new findings, providing a more comprehensive analysis of the data and addressing the gaps mentioned by the reviewer, thereby strengthening the manuscript’s scientific contribution.

Round 3

Reviewer 1 Report

Comments and Suggestions for Authors

I am happy with the current form of the manuscript, and it is ready to be accepted for publication.